# All-Atom Molecular Dynamics Simulations of Dimeric Lung Surfactant Protein B in Lipid Multilayers

**DOI:** 10.3390/ijms20163863

**Published:** 2019-08-08

**Authors:** Nicholas A. S. Robichaud, Mohammad Hassan Khatami, Ivan Saika-Voivod, Valerie Booth

**Affiliations:** 1Department of Physics and Physical Oceanography, Memorial University of Newfoundland, St. John’s, NL A1B 3X7, Canada; 2Department of Biochemistry, Memorial University of Newfoundland, St. John’s, NL A1B 3X9, Canada

**Keywords:** lung surfactant, surfactant protein B, SP-B, protein structure, lipid, membrane, molecular dynamics, simulation, all-atom

## Abstract

Although lung surfactant protein B (SP-B) is an essential protein that plays a crucial role in breathing, the details of its structure and mechanism are not well understood. SP-B forms covalent homodimers, and in this work we use all-atom molecular dynamics simulations to study dimeric SP-B’s structure and its behavior in promoting lipid structural transitions. Four initial system configurations were constructed based on current knowledge of SP-B’s structure and mechanism, and the protein maintained a helicity consistent with experiment in all systems. Several SP-B-induced lipid reorganization behaviors were observed, and regions of the protein particularly important for these activities included SP-B’s “central loop” and “hinge” regions. SP-B dimers with one subunit initially positioned in each of two adjacent bilayers appeared to promote close contact between two bilayers. When both subunits were initially positioned in the same bilayer, SP-B induced the formation of a defect in the bilayer, with water penetrating into the centre of the bilayer. Similarly, dimeric SP-B showed a propensity to interact with preformed interpores in the bilayer. SP-B dimers also promoted bilayer thinning and creasing. This work fleshes out the atomistic details of the dimeric SP-B structures and SP-B/lipid interactions that underlie SP-B’s essential functions.

## 1. Introduction

Lung surfactant (LS) is a complex mixture of lipids and proteins that is essential to life [1,2,3,4,5,6,7,8]. Its critical contributions arise from its ability to lower the surface tension at the air/water interface in the lung’s alveoli, reducing the work of breathing and preventing alveolar collapse. LS forms a lipid monolayer at the air/water interface, with the lipid acyl chains exposed to air, as well as multilayer structures in the aqueous hypophase. Connections between the monolayer and multilayers are critical to function throughout the breathing cycle, which entails an expansion of the surface during inhalation, and contraction during exhalation. Both processes involve the transfer of LS material between the monolayer and the multilayer reservoir, and the hydrophobic LS proteins are critical to these structural transformations.

Both the lipid and protein compositions of LS are unique with respect to other membranes found in the body. The lipid component is highly enriched in dipalmitoylphosphatidylcholine (DPPC), a zwitterionic phospholipid with two saturated acyl chains that is capable of forming a tightly packed structure. This capacity for tight packing means that DPPC can withstand the surface tension at the air/water interface, helping to resist alveolar collapse during expiration [9]. LS lipids also include unsaturated PC lipids, cholesterol, as well as anionic lipids such as phosphatidylglycerols, that are thought to be important in interactions with lung surfactant proteins. The protein complement of LS includes two hydrophilic proteins in the collectin family, surfactant proteins A and D (SP-A and SP-D) [10,11,12], which make important contributions to the lung’s immune defenses against pathogens. Crucial are the two small hydrophobic proteins, SP-B and SP-C [13,14,15], which stabilize the lipid structures and transitions critical to LS function. SP-B itself is vital to lung function, as its absence is fatal in humans with genetic disorders [16], as well as in SP-B-knockout mice [17]. The absence of SP-C is not fatal, although its loss does lead to lung mechanics abnormalities [18].

Premature infants frequently develop respiratory distress (RDS) if they are born before their lungs produce a sufficient quantity of lung surfactant [6,19]. The advent in the 1990s of animal-derived lung surfactants that included the hydrophobic proteins SP-B and SP-C greatly improved the survival of premature infants [20,21]. However, this success story has not been repeated in attempts to use exogenous lung surfactant treatments for other respiratory disorders in non-neonates, for example with acute respiratory distress syndrome (ARDS) [22]. A major challenge has been the cost of producing animal-derived lung surfactant for larger patients who require much more LS, especially if—as in ARDS—there are LS-inactivating conditions in the lungs which may require larger and repeated doses of LS. Furthermore, the peculiar characteristics of SP-B and SP-C (e.g., their extreme hydrophobic nature) have made it difficult to produce these critical protein components of LS in the lab. These constraints have meant that almost all clinical trials for exogenous LS treatment in non-neonates have contained no SP-B or SP-C. Lipid-only exogenous LS trials have not gone well, which is unsurprising given that SP-B is an essential protein [22]. Furthermore, the one clinical trial with animal-derived SP-B and SP-C was not successful [23], likely due to insufficient alveolar delivery of the exogenous LS [24]. Hence, in recent years there has been an intense focus on designing peptides that mimic the function of SP-B and SP-C [25,26,27]. Of course, an understanding of the structure–function relationships that underlie the proteins’ contributions to lung function is also of critical importance to the rational design of such peptides.

There has been intense interest in determining the atomic-resolution structures of SP-B and SP-C, both from the point-of-view of understanding the basic science of how these proteins work, as well as from the perspective of producing effective SP-B- and SP-C-mimetic peptides for inclusion in exogenous lung surfactant therapeutics. SP-C is a 35-residue peptide that is naturally palmitoylated at cysteines 5 and 6, and forms a single transmembrane helix [13,28]. SP-B, on the other hand, does not have an experimental structure, which can largely be attributed to its larger size combined with its extreme hydrophobicity [29], and the consequent difficulties in expressing it recombinantly and determining its structure by standard structure determination techniques such as NMR and crystallography. SP-B is 79 residues in length, and is a member of the saposin superfamily. Based on its homology to known structures, it is believed to form four to five amphipathic helices [29,30]. As well as their helicity, saposin superfamily proteins share six cysteine residues that form intramolecular disulfide bonds that help stabilize the interactions between pairs of helices (Figure 1). However, it is uncertain if SP-B forms a closed, globular structure, with the helix pairs packed to minimize the surface-exposed hydrophobic residues, as seen for some saposin superfamily proteins such as NK-lysin ([31], PDBID 1NKL). Alternatively, SP-B might form a much more open structure, as seen for saposin A ([32], PDBID 4DDJ) and saposin C ([33], PDBID 1SN6), which would provide a large hydrophobic surface that SP-B could use to interact with lipids. It is even possible that SP-B’s topology changes during the respiratory cycle (i.e., as the air/water surface is compressed upon expiration and expands upon inspiration), or that the SP-B topology is different in different LS lipid structures. One additional structural feature to note is the seven N-terminal residues of SP-B, with sequence FPIPLPY, which have been termed the “insertion sequence” and have been proposed to help SP-B insert into the lipid monolayer at the air/water interface [34,35].

The structure of SP-B and its consequent effects on LS lipid structures are clearly tied to its mechanism of action. SP-B has been proposed to contribute to several types of lipid structures and structural transformations that are thought to be key to LS’ mechanism. Firstly, SP-B is critical in promoting the fast adsorption of LS to the air/water interface, likely through the stabilization of highly curved hemifusion-like structures that are necessary intermediates to the transfer of LS material from the aqueous phase to the air/water interface [36,37]. Secondly, SP-B appears to tie the multilayer stacks of LS together strongly when the surface area of the interface drops upon expiration [35,38]. Such stacks could increase the surface film’s strength and resistance to collapse. Thirdly, SP-B has been shown to be necessary for the organization of the multilayer structures that are driven out of the interface upon compression such that they can be efficiently re-incorporated in the surface film upon expansion (i.e., LS “refinement”) [37,39].

Even the oligomeric state of SP-B responsible for its biological function is something of an open question. Unlike other saposin superfamily proteins, SP-B contains a seventh cysteine residue which stabilizes the dimers that have traditionally been considered the functional unit of SP-B [14]. However, rodents with a mutant form of SP-B that lacks this seventh cysteine do not show major impairments [41]. This may imply that dimerization is not important for function, that non-covalent interactions are capable of sufficiently stabilizing the dimers, or that SP-C alone is sufficient to support the respiration of these mice. Depending on which detergents/solvents are used to extract and purify native SP-B from animal lungs, SP-B has also been observed to form large oligomeric ring-like structures [42], which may be the form SP-B takes on for some or all of its functional mechanisms.

Recent computer simulation studies of SP-B lipid interactions have included the coarse-grained simulation work carried out by Baoukina and co-workers [43,44,45], as well as all-atom simulation studies carried out previously in our group [40] and all-atom simulations of peptide fragments of SP-B [46]. The coarse-grained simulations [43] have shown very interesting behaviour, with SP-B promoting fusion between disconnected monolayers and bilayers, as well as promoting the local bending of lipid monolayers. The latter behaviour was observed for aggregates of SP-B and had faster kinetics if SP-B was covalently dimerized. On the other hand, all-atom molecular dynamics of monomeric SP-B [40] revealed a number of details of SP-B structure and lipid interactions. In addition to fine-tuning the homology-based structural model of SP-B, the all-atom simulations of SP-B monomers showed that SP-B is able to take on a variety of energetically feasible topologies within the bilayer, including with its helices in the plane of the bilayer or in a more transmembrane orientation. SP-B was also seen to promote and stabilize defects in the lipid bilayer, lipid re-organization abilities that apparently derive from several features including the structural plasticity in its central loop region, as well as bending at the “hinge” between the two pairs of helices (Figure 1).

The current work extends our previous study in two ways: (1) to dimeric SP-B; and (2) to multilayer lipid structures. Final structures from the previous monomer simulations were used to create dimers connected by the disulfide bond at C48. These dimers were used as initial structures in systems with one or two bilayers.

## 2. Results

Four different systems with SP-B dimerized via the native disulfide bond were constructed and then subjected to all-atom molecular dynamics (Table 1). The initial configurations were based on the final structures from previous simulations of monomeric SP-B [40], as detailed in the Methods section. Briefly, two of the systems contained one POPC lipid bilayer with dimeric SP-B embedded in the lipids in one of two initial configurations, bent (Figure 2a) or open (Figure 2b), termed Bent In One Bilayer (BI1), and Open In One Bilayer (OI1), respectively. Another two simulations were composed of two POPC lipid bilayers, with dimeric SP-B initially positioned between the bilayers, with SP-B in a closed (Figure 2c) or open (Figure 2d) conformation, termed Closed Out Two Bilayers (CO2) and Open In Two Bilayers (OI2), respectively. All bilayers were set up with a bilayer pore, distant from the protein, that provided a means for the bilayers to relax and relieve any stress due to differences in lipid area.

In addition to representing well-energy-minimized structures from previous monomeric simulations, the chosen starting configurations also represent classes of SP-B structures that have been proposed to account for SP-B’s activity, as follows. The closed configuration was included to match early homology models of SP-B that were based on closed saposin superfamily structures determined in water [30]. The open configurations were inspired by the structure envisioned by many experimentalists whose work probes the functional mechanisms of SP-B (e.g., [3,5]). The bent configurations represent an intermediate topology between these open and closed configurations that seemed particularly active in promoting bilayer structure transitions in earlier simulations of the SP-B monomers [40].

The simulations equilibrated slowly, as expected for such large protein/lipid systems (Figure 3). Despite the long simulation times, none of the simulations at 310 K reached a convincing steady state in potential energy, although the single membrane simulations, BI1 and OI1, may be approaching a plateau (Figure 3a). The overall rate of potential energy decrease was gradual in every simulation, suggesting that these four configurations of the protein–membrane system are all energetically feasible.

Root mean square deviation (RMSD) traces the conformational stability of the protein in each simulation (Figure 4a). The range of RMSD values was between approximately 0.2 and 0.7 nm, with the OI1 conformation and one subunit of the CO2 conformation showing the highest degree of structural fluctuations and BI1 and OI2 the lowest. In all four simulations SP-B maintained an overall level of helicity (Figure 4b) consistent with experimentally determined values, between 30% and 40% [29]. The helicity decreased only moderately from the initial value, indicating that the covalent dimerization does not have a large impact on SP-B’s secondary structure. Overall, as the simulations progressed, the number of protein-lipid hydrogen bonds increased for all systems, and the number of protein-protein hydrogen bonds stayed roughly constant in the single bilayer systems and decreased in the two bilayer systems (Appendix A).

The CO2 simulation showed no changes in bilayer structure after 1.5 μs and was thus terminated at that point. However, the remaining three simulations showed interesting alterations to the bilayer structures. Furthermore, since the OI2 simulation showed the beginnings of close bilayer-bilayer contact after 3 μs, the temperature was increased from 310 to 340 K to help overcome energetic barriers and run for an additional 4.2 μs to further explore SP-B-induced bilayer transitions. Only the OI2 simulation was chosen for the high temperature segment as, judging from potential energy (Figure 3b), it was the furthest from equilibrium, and was the only simulation that appeared to be potentially on the cusp of a structural transition at the end of the 310K run.

### 2.1. Bent In One Bilayer (BI1)

In the BI1 system, the SP-B dimer was initially located within the acyl chain region of a single bilayer, with SP-B in a bent configuration. SP-B’s N- and C-terminal helices (Figure 1) started in contact with the headgroups of the upper leaflet, parallel to the plane of the membrane (Figure 2a). SP-B’s central loops, however, started in contact with lipid headgroups in the opposing leaflet (i.e., the lower one). The helical secondary structure of the original structures was largely preserved throughout the simulation, with the longest helices adjacent to the N- and C-termini and small sections of helical structure elsewhere (Figure 5a). The two subunits maintained a very similar secondary structure to each other.

Interestingly, in terms of SP-B-promoted lipid reorganization, during the simulation the two subunits moved closer together (Figure 6a) and substantially deformed the bilayer. In particular, SP-B’s central loop and adjacent regions from each subunit approached each other. Together, the two subunits stabilized a sizeable defect in the bilayer, with lipid headgroups and water entering deeply into the acyl chain region. In the defect, water molecules formed a continuous chain from one side of the bilayer to the other. Water in the defect was very mobile; it frequently entered and left again. This lipid defect appeared early in the simulation and persisted until the very end, suggesting it is a stable lipid/protein configuration.

It is also interesting that later in the same simulation, the N-terminal insertion sequence (residues 1–7) and the N/C-terminal helix pair of one of the subunits came into contact with the preformed pore in the bilayer and transitioned from a membrane-parallel to a highly tilted orientation that lined the pore (Figure 6b). The overall structure for this subunit (blue in Figure 6b) that persisted throughout the last 0.5 μs was an “arc” shape. The central region of the arc was located at the polar/apolar interface of the upper leaflet, roughly parallel to the membrane, and with low helicity. By contrast, the two ends of the arc took on a transmembrane orientation that reached into both leaflets and, especially in the case of the N- and C-terminal region, had high helicity (Figure 5a and Figure 6b). The central loop and adjacent region stabilized the SP-B-induced bilayer defect, while the N/C-terminal helix pair stabilized the preformed pore.

The trajectories were carefully examined for any interactions (e.g., salt bridges) that, in addition to the disulfide, might form between SP-B’s subunits. Apart from a few very short-lived inter-subunit hydrogen bonds (seen in <0.5% of frames), there appeared to be no non-covalent interactions between the two chains. It may be that since this region of both subunits has energetically favourable interactions with the lipid defect, the defect itself helps keep the two chains in close association with each other. That is to say, with the exception of the C48 disulfide, the interaction between the subunits was not direct but was mediated by the lipids.

The region of membrane close to the central loop region of SP-B was substantially thinner than the membrane as a whole. While most of the membrane was ~4.3 nm across from phosphate to phosphate, the membrane near C48—the residue that stabilized the dimer—was only 3.5 nm across (Figure 7a). Both hydrophobic and electrostatic interactions appeared to have a role in stabilizing the SP-B-promoted defect in the bilayer (Figure 7b). As is frequently the case for arginine in membrane proteins, SP-B’s arginine residues appeared to provide anchoring to the interface. For example, in both subunits, R52 (orange balls/sticks in Figure 7b) was found in the upper leaflet in close proximity to the lipid headgroups (Figure 7c) and, at any one time, each had an average of approximately two hydrogen bonds to the lipid headgroups (Figure 7d). Similar protein–lipid interactions were observed for the other charged amino acids in both subunits, as confirmed by radial distribution function calculations (Appendix A). Notable exceptions were the R36 residues, which were located deep in the bilayer, away from the lipid headgroups, and consistently formed hydrogen bonds with the water permeating through the bilayer defect (Appendix A). R17 and the adjacent K16 both conspicuously anchored the side of the SP-B arc near the preformed pore.

Considering the lower leaflet of the bilayer, the SP-B region that made the most lipid contacts was the loop from residue 37 to 45, with the sequence VVPLVAGGI (green balls/sticks in Figure 7b). The position of one residue within the loop, P39, was examined as an indicator of the position of the loop over time and was found to be located about midway between the centre of the bilayer and the headgroups of the lower leaflet throughout the simulation (Figure 7a). The 37 to 45 loop contains several hydrophobic residues: three valines, an alanine, and an isoleucine. It has little regular secondary structure (Figure 5a) and has a glycine pair that is likely to be conformationally dynamic. Unlike with the upper leaflet, there were few hydrogen bonds between this central loop and lipid headgroups (Figure 7e,f), although there were several between the protein backbone and water, in keeping with the close proximity of the water molecules that penetrated into the bilayer in the region of the SP-B-induced defect. The combination of a disulfide bridge and a lack of direct physical bonding between the two subunits near the lower leaflet allowed the defect to be broad (Figure 7b) relative to the defect induced by a single monomer [40]. Were the loop regions of the two subunits to bind more tightly to each other, the defect would be narrower.

### 2.2. Open In One Bilayer (OI1)

In contrast to the BI1 system, the OI1 system started with all the helices of SP-B in parallel to the interface (Figure 2b), in contact with only one leaflet of the bilayer. Overall, the OI1 system maintained a similar degree of helicity to the BI1 system (Figure 4b). Interestingly, as the system evolved, a sizable crease in the SP-B-containing leaflet formed (Figure 8a,b). The crease formed along the long-axis of SP-B and was accompanied by drastic thinning of the SP-B-containing leaflet, which allowed the hydrophobic face of SP-B to make contact with the acyl chains on the opposing bilayer leaflet. The SP-B-induced bilayer thinning can be seen in Figure 8c, as the region of the membrane close to the protein was ~0.5 nm thinner than the rest of the membrane. The protein backbone remained deeply embedded within the acyl chain region of the bilayer throughout the simulation (Figure 8c).

Approximately 2 μs into the simulation, the N/C-terminal helix pair of one SP-B subunit encountered the preformed pore in the membrane. This contact occurred well after the crease induced by SP-B had completely formed. As for BI1, the portion of SP-B that interacted with the preformed pore was the N/C-terminal helix pair, which tilted to about 45° to line the side of the pore. A pair of positively charged residues (i.e., K16 and R17) were located at the crux of this bend in the protein, tightly interacting with the phosphate groups of the lipids in the preformed pore. Also reminiscent of the BI1 system, in the OI1 system, the central loop region of SP-B probed the furthest towards the opposing leaflet of the bilayer (red subunit in Figure 8b,d). The hydrogen bonding pattern was similar to that seen for BI1, that is, no hydrogen bonds between the protein subunits and many protein–lipid hydrogen bonds (see Appendix A). Throughout the simulation, most charged residues of SP-B interacted closely with lipid headgroups, as confirmed by radial distribution function calculations (Appendix A), or were involved in intra-chain salt bridges in a pattern consistent with findings from previous monomer simulations [40].

### 2.3. Closed Out Two Bilayers (CO2)

In this system, an SP-B dimer, with each subunit initially in a closed conformation, was placed in the aqueous layer between two bilayers (Figure 2c). The protein’s structure evolved relatively quickly, settling in to its final structure by ~700 ns, with many of the hydrophobic residues partitioned in the compact protein interior away from water (Figure 4a). Some hydrophobic sidechains on one subunit—particularly those in the central loop—made hydrophobic contacts with the other subunit, notably to the LAVAV sequence starting at residue 27, as well as to tryptophan 9 and leucine 10. There was no significant hydrogen bonding or salt bridging between subunits.

There was no sign of any protein-promoted bilayer–bilayer close contact or lipid re-organization (Figure 9a), and thus the simulation was terminated at 1.5 μs. The final SP-B structure had a helicity of ~30% (Figure 4b). As for the other systems, there were extensive interactions between the charged sidechains of each SP-B subunit with the lipid headgroups. Radial distribution function calculations revealed close interactions between the positively charged arginine and lysine sidechains and the negatively charged phosphate groups of POPC from both bilayers. However, of the 14 total arginines from both subunits, five showed little to no interaction with the lipids, and were instead sequestered within the water layer. Of the two negatively charged residues in SP-B, the E51 sidechains tended to form transient interactions with the adjacent R52 on the same chain, and neither E51 nor D59 sidechains had significant interaction with the bilayers. Given SP-B’s positioning outside the bilayers in this system, the interactions of the positively charged sidechains with the lipid headgroups served to maintain close contact between the two bilayers.

Although the amino group F1 of SP-B made contact with the lipid headgroups (green lines in Figure 9b), residues 2–7 of SP-B’s insertion sequence did not appear to insert into the headgroup region as the simulation progressed. Over the course of the simulation, the distance between the two bilayers shortened very slightly (Figure 9a,b).

### 2.4. Open In Two Bilayers (OI2)

The final system was set up with two bilayers and two subunits of SP-B in an open conformation. The subunits were placed asymmetrically in the headgroup region of different bilayers (Figure 2d). During the first 2.5 μs of simulation, one of the SP-B subunits had travelled partially out of its initial bilayer and partially into the other bilayer, led by its central loop region (Figure 10a). As this transition occurred, the protein appeared to carry some lipids partway with it such that there was very close contact between the headgroups of the two bilayers, although no lipids were exchanged between the bilayers.

There did not appear to be any additional SP-B-promoted bilayer–bilayer contacts after a total of 3 μs of simulation, and so at this point the temperature was increased from 310 to 340 K, and the simulation ran for an additional 4 μs at the higher temperature. With the increase in temperature, SP-B adjusted into a more symmetrical orientation with respect to the bilayers (Figure 10b), with the C48 disulfide localized near to the midpoint of the water layer between the bilayers (Figure 10b,c). Intriguingly, in the final structure, the central loop of one of the subunits was fully embedded in the opposing bilayer (blue subunit in Figure 10b). Residues 33 to 37 (QVCRV) and residues 43 to 52 (GGICQCLAER) of this subunit were stationed within the water layer or at the lipid/water interface for more than 90% of the simulation (including both the lower- and higher-temperature segments).

In terms of inter-subunit contacts, in the high-temperature portion of the OI2 simulation, a hydrogen bond was detected between E51 on one subunit to R52′ on the other subunit in about 7% of the ~2100 frames analyzed. At this high temperature, the largely hydrophobic central loops of the monomers were in close contact. Inter-chain backbone hydrogen bonds between these central loops were also present, most prominently between G43 and I45′, which was observed in ~60% of the frames, and between A42 and G43′, which was observed in ~30% of the frames. None of these inter-chain interactions were detected before temperature was raised.

## 3. Discussion

A variety of structural features of SP-B have previously been suggested to contribute to SP-B-mediated lipid structural transitions, and will be considered in turn in the light of this work’s simulations of SP-B dimers. Firstly, the N-terminal seven residues of SP-B, with sequence FPIPLPY, have been termed the “insertion sequence” and have been proposed to be important in helping SP-B insert into the air/lipid interface [34,35]. In all simulations, with the exception of the CO2 system, this region of SP-B was observed to localize tightly to the polar/apolar interface of the lipids. The insertion sequence did not contact the deeper, more hydrophobic, region of the bilayers, in keeping with the amphipathic character of its chemical structure. In two of the three simulations where it inserted, the insertion sequence remained in the planar part of the polar/apolar interface. However, in the BI1 simulation it made the initial contact with the preformed pore, at which point it rapidly translocated along the pore, to the other side of the bilayer (Figure 6b).

Secondly, in simulations of monomeric SP-B [40], the “hinge” region between the two pairs of helices (Figure 1) appeared to figure importantly in allowing the structural flexibility for SP-B to interact with curved lipid structures, including both the preformed lipid pores and the SP-B-induced defect [40]. The same appeared to be true for the dimer structure, with the hinge bending to allow a variety of SP-B–non-planar-bilayer structural complexes (e.g., Figure 6b and Figure 8d). The “bent” conformation of monomeric SP-B was stabilized by a key intra-chain salt bridge from K24 to D59, and the same appeared to be true for the dimer (Appendix A).

Thirdly, the loop that connects SP-B’s two pairs of helices, which we term the “central loop” (Figure 1), also came to our attention initially due to its behavior in the SP-B monomer simulations [40]. This loop, with sequence VVPLVAGGI, appeared to be particularly dynamic in the SP-B monomers and was often seen to “probe” into relatively distant portions of the lipid structures. The same appeared to hold true for dimeric SP-B. In the BI1 system, the central loop was apparently the key feature in inducing the bilayer defect. It was structurally dynamic and formed backbone hydrogen bonds with the water molecules that penetrated into the disordered lipid region (Figure 7). Likewise, in the OI1 system, the central loop was the only part of SP-B that made substantial contact with the opposite leaflet of the bilayer (red subunit in Figure 8). Furthermore, in the OI2 simulation, the central loop of the blue subunit intriguingly explored into the upper bilayer, while the bulk of the subunit remained in the lower bilayer (Figure 10b). The central loop sequence is highly conserved (e.g., almost identical in SP-B from a variety of mammals). Overall, this central loop region appears to have a special role in promoting lipid disorder and hence presumably in functional lipid structural transitions. It may thus be important to include the central loop sequence in SP-B-based therapeutic peptides. Interestingly, a highly functional fragment of SP-B termed “Mini-B” does contain a GG pair in the loop between its two helices, although the rest of the loop sequence is different [46,48]. It would be interesting to see if adding the full central loop region identified here to Mini-B would alter its function.

Fourthly, while the biologically functional oligomeric form of SP-B has not been firmly established, it is generally accepted to dimerize via C48, with possible contributions to dimer stabilization from salt bridges, especially E51 on one subunit to R52 on the other [30,42]. However, this salt bridge was relatively rare in our dimer simulations. In most cases, E51 and R52 were hydrogen-bonded to lipid head groups and not to each other. The most E51 to R52 bonding observed was for the OI2 system, where it was seen in 7% of the high temperature frames. On the other hand, having only a single point of contact between the subunits at C48 appeared to allow the dimer to stabilize a large variety of lipid structures. For example, in the SP-B-induced lipid defect in BI1 (Figure 7b), C48 was the only point of contact between the two subunits, which allowed both neighboring central loops the conformational freedom key to promote the bilayer defect.

Fifthly, the large number of positively charged residues in SP-B, which have long been viewed as critical to SP-B’s lipid interactions, appear to provide points of strong anchoring to the polar/apolar interface. These anchors may provide a balance to the conformational flexibility of the central loop, C48, and hinge regions. On the one hand, the flexible regions promote local lipid disorder and membrane thinning (e.g., in the lower leaflet of Figure 7b). On the other hand, the anchoring interactions with lipid headgroups from the top leaflet stabilized the bent transmembrane configuration of SP-B, allowing the central loops to integrate themselves into the acyl chain region and promote the formation of a sizable defect over time. As the protein probes even deeper into the lower leaflet, the ionic interactions with the upper leaflet might also allow the protein to pull in lipid headgroups from the top leaflet, further encouraging bilayer thinning. Hence, the combination of the interface anchoring with the bilayer disordering may underlie some of SP-B’s lipid reorganizing mechanisms.

Regarding the large crease formed in the OI1 simulation, we note that the depth to which the dimer descended into the membrane was approximately the same as for the monomeric protein [40]. However, the depth should be sufficient to allow OI1-type dimers (or even monomeric proteins) in opposing leaflets to interact. We hope to explore this interaction in future investigations.

While in this work only lipids with zwitterionic headgroups were employed—LS has 8%–15% anionic lipids [39]. Thus, the SP-B–lipid interactions observed in these simulations are likely to become even stronger in the presence of anionic lipid, which is something to explore in future work.

In considering how these all-atom results compare with earlier coarse-grained simulations [43,44,45], it is good to bear in mind the longer timescales of the coarse-grained work, which of course come at the cost of reduced structural detail. Nonetheless, the close bilayer–bilayer contacts encouraged by SP-B in the OI2 system (e.g., Figure 10b,c) did appear to resemble the early stages SP-B-promoted vesicle and bilayer/monolayer fusion activities observed in the coarse-grained simulations, where bent SP-B forms a “scaffold” that promotes the formation of a lipid bridge. Curiously, the SP-B-promoted bilayer crease observed in the bilayers here (Figure 8b) entailed development of the opposite lipid curvature than observed for the coarse-grained simulations of SP-B interacting with monolayers. That is, in the all-atom simulations, SP-B promoted negative curvature, with headgroups bending inwards, while in the coarse-grained simulations the curvature was positive. It is also worth noting that the kinetics of fold formation in the coarse-grained systems were seen to be faster with dimeric SP-B than with monomeric SP-B, which is in keeping with how early dimeric SP-B induced crease formation in the all-atom simulation (Figure 8b).

## 4. Materials and Methods

The initial SP-B dimer structures (Figure 2) were constructed from the monomeric SP-B configurations captured at—or nearly at—the end of our previous SP-B simulations [40]. These monomeric configurations had already been subjected to extensive simulation time within the lipid environment, and thus constituted energetically feasible starting points for the dimer simulations. The SP-B monomer structures were modified to create SP-B dimers by connecting them via a C48-to-C48′ disulfide bond. Three of the six final configurations from the earlier simulation provided C48 configurations that were compatible with forming structurally feasible inter-chain disulfides. The disulfides were formed manually with Swiss pdb Viewer [49] without any modifications (e.g., extra local relaxation).

In constructing the new SP-B systems, care was taken to preserve—as much as possible—the lipid environment around the protein structures. The “Bent In One Bilayer” (BI1) system (lycero-3-phosphocholine (POPC). Both the “Open In One Bilayer” (OI1, Figure 2a) was produced by combining two identical “bent-in” monomers (see [40] for the nomenclature of the monomeric systems) and the “Closed Out Two Bilayers” (CO2) by combining two identical “bent-out” monomers (Figure 2c). The lipid component of the system was 1-palmitoyl-2-oleoyl-sn-g (Figure 2b) and “Open In Two Bilayers” (OI2, Figure 2d) systems were generated by combining two identical “open-in” monomers. For the “Open In One Bilayer” system, the two subunits, in extended form, were placed in the same leaflet (Figure 2b). In contrast, for the “Open In Two Bilayers” system the protein subunits were placed in opposing leaflets of two separate bilayers (Figure 2d) with one subunit (in red) kept in its original lipid environment, while the other subunit (in blue) was manually placed among the head groups of a second lipid bilayer at a shallow angle. Any lipids that were overlapping with the protein were removed. Preformed pores in the lipid bilayers allowed the lipid molecules to relax to an appropriate lipid–area ratio [40,50]. The pores also are advantageous in providing a curved bilayer structure that, although not initially positioned close to the protein, could be encountered by the protein later on in the simulation, potentially revealing SP-B structures that energetically favour interacting with such a lipid conformation. A snapshot of a preformed pore is shown in Appendix A.

After preparing the initial protein–lipid configurations, we used GROMACS 5.1, employing the OPLS-AA force field [51,52] to run the molecular dynamics (MD) simulations. The system was solvated using TIP4P water molecules and sufficient Cl^−^ added to make the system neutral (Table 1). After energy minimization under NVT conditions, each system ran for 2 ns at T = 310 K with the protein position restrained and a time step of 2 fs. In the production run, the systems were simulated under NPT conditions at 310 K and 1 atm, using Parrinello–Rahman semi-isotropic pressure coupling, with τ_p_ = 5 ps and compressibility of 4.5 × 10^−5^ bar^−1^. During the production run, no position or structural restraints were placed on the protein. The time step of the simulations was set to 2 fs and they ran for 1.5 to 3 μs (as detailed in Table 1). For the single-bilayer simulations, the cells were approximately 12 nm × 12 nm × 8.5 nm, and for the two-bilayer simulations the cells were approximately 11 nm × 11 nm × 12.5 nm and 11 nm × 11 nm × 14.5 nm for OI2 and CO2, respectively. After 3 μs, the temperature of the OI2 system was increased to 340 K and run for an additional 4.3 μs.

Analysis of the simulations was performed primarily using Gromacs tools and Visual Molecular Dynamics software (VMD) [53]. Calculations of helicity were performed using STRIDE [47]. A residue was considered helical if STRIDE labelled it as in an α-helix, 3_10_ helix, or π-helix. This data was then used to calculate the percentage of helical residues at each time step and the time-averaged helicity per residue (as depicted in Figure 4b and Figure 5, respectively). Coordinate plots illustrating bilayer thickness or bilayer separation for one and two bilayer systems, respectively, were created by using the Gromacs tool “gmx density” to calculate positions of maximum density for phosphorus atoms in each leaflet of bilayers. In performing this analysis, lipid headgroups positioned near the protein were singled out at each frame to determine local effects of the protein on bilayer structure. Due to the distinct protein configurations in these four systems, this procedure differed for each simulation, as noted in the figure captions. For the OI2 system, lipid headgroups situated near where SP-B crosses through the central water layer were identified in the following way: First, two representative residues, R36 and C46 on one subunit (blue in Figure 2d), were chosen. These residues were continually located within the water layer on opposing sides of the central loop in the protein. Then, at every frame, all protein residues found within 1 nm of both R36 and C46 were identified. The two residues were chosen such that this selection consistently defined the region of SP-B within the water layer and at the lipid/water interface. Finally, the phosphorus atoms in lipid headgroups that were located within 1 nm of lateral distance from this selected portion of the protein were identified at every frame and used for density calculations and plotting, as depicted in Figure 10.

## 5. Conclusions

Dimeric SP-B maintained a helicity of 27–35% in all the systems, which is consistent with experiment [29], and demonstrated a variety of other structural features important for promoting non-planar lipid structures, including the probing central loop and the flexible hinge between the two pairs of helices (Figure 1). Knowledge of these structural features will be helpful in the design of SP-B-mimetic therapeutic peptides.

Turning to lipid interactions, SP-B dimers with one subunit positioned in each of two adjacent bilayers appeared to promote close contact between two bilayers (Figure 10b,c), in keeping with suggestions that one of SP-B’s mechanisms is in LS “refinement”. SP-B has been shown to organize the lipid bilayers driven out during expiration, in such a way that the lipids can be efficiently re-incorporated into the surface film upon inspiration [37,39]. When both SP-B subunits were initially positioned in the same bilayer, SP-B dimers appeared to promote several mechanistically relevant lipid structural transitions: (1) substantial bilayer thinning (Figure 7a and Figure 8c); (2) the formation of a defect with water and lipid headgroups entering into the centre of the bilayer (Figure 6b); (3) the stabilization of preformed lipid pores, as evidenced by the persistence of the SP-B/pore lipid structure (Figure 6b and Figure 8d); and (4) the formation of a substantial bilayer crease (Figure 8b). These behaviors may support not only SP-B’s lipid-refining capabilities, but also SP-B’s well-established activities in promoting adsorption to the surface film, which must involve the formation of non-planar lipid structures. We and others [14,30,39,40,45] have envisioned the various SP-B-promoted lipid structures as potentially unified by the SP-B promotion of lipid structures analogous to the standard structural steps in membrane fusion. As detailed in [40], the key SP-B-promoted lipid structures would be (1) close contact of the two membranes (Figure 10b), (2) thinning (Figure 7a and Figure 8c) and hemifusion stalk formation, and (3) fusion pore formation (Figure 6b). Overall, this work fleshes out the atomistic details of the dimeric SP-B structures and SP-B/lipid interactions that underlie SP-B’s essential biological functions.

## Figures and Tables

**Figure 1 ijms-20-03863-f001:**
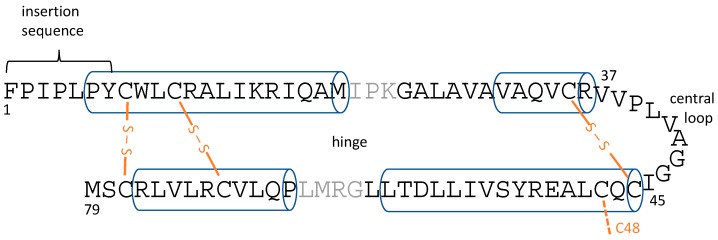
Overall topology of lung surfactant protein B (SP-B) helices and disulfide bonds with helical regions shown as cylinders and disulfide bonds in orange. C48 is thought to stabilize SP-B homo-dimers. Earlier simulation studies [40] pointed to the role of the central loop and hinge region flexibility in stabilizing bilayer defects. Note that the position of the helices varies slightly between simulations.

**Figure 2 ijms-20-03863-f002:**
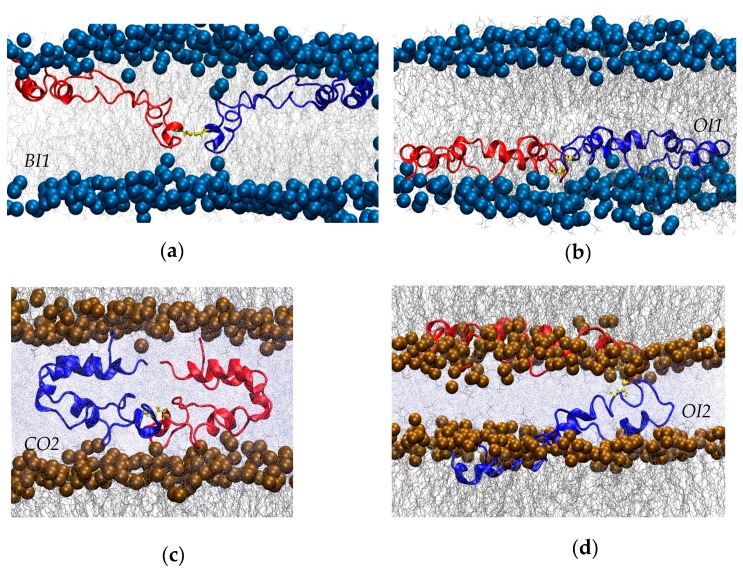
Close-ups of initial conformations of the four simulations. (**a**) “Bent In One Bilayer” (BI1). (**b**) “Open In One Bilayer” (OI1). (**c**) “Closed Out Two Bilayers” (CO2). (**d**) “Open In Two Bilayers” (OI2). The inter-subunit disulfide is shown in yellow, the phosphorus atoms in the lipid headgroups as solid spheres, lipid chains in grey, and water (only shown in panels (**c**,**d**) in pale blue.

**Figure 3 ijms-20-03863-f003:**
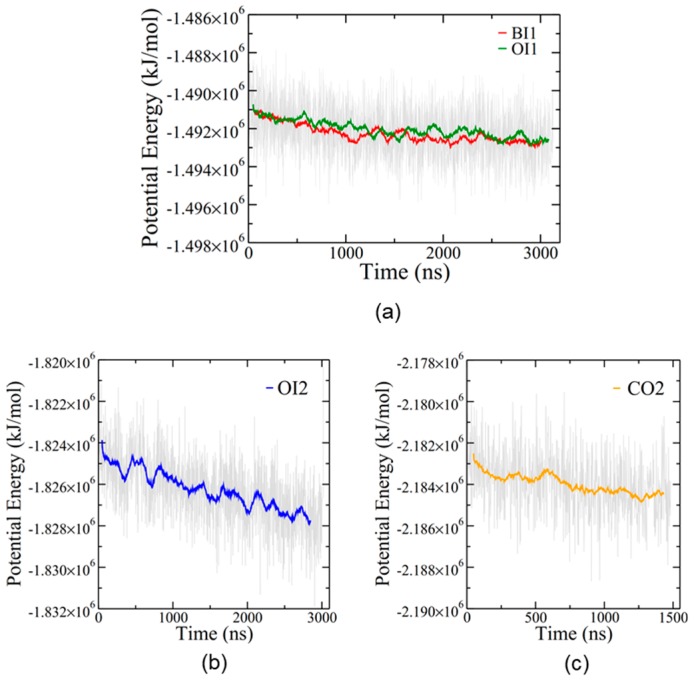
Potential energy for the four simulations. (**a**) BI1 (red) and OI1 (green). (**b**) The first 3 μs of OI2 (before the temperature increase). The full potential energy plot for OI2 is shown in Appendix A. (**c**) CO2. Coloured lines represent running averages over 50 points while grey indicates individual time steps spaced 2 ns apart.

**Figure 4 ijms-20-03863-f004:**
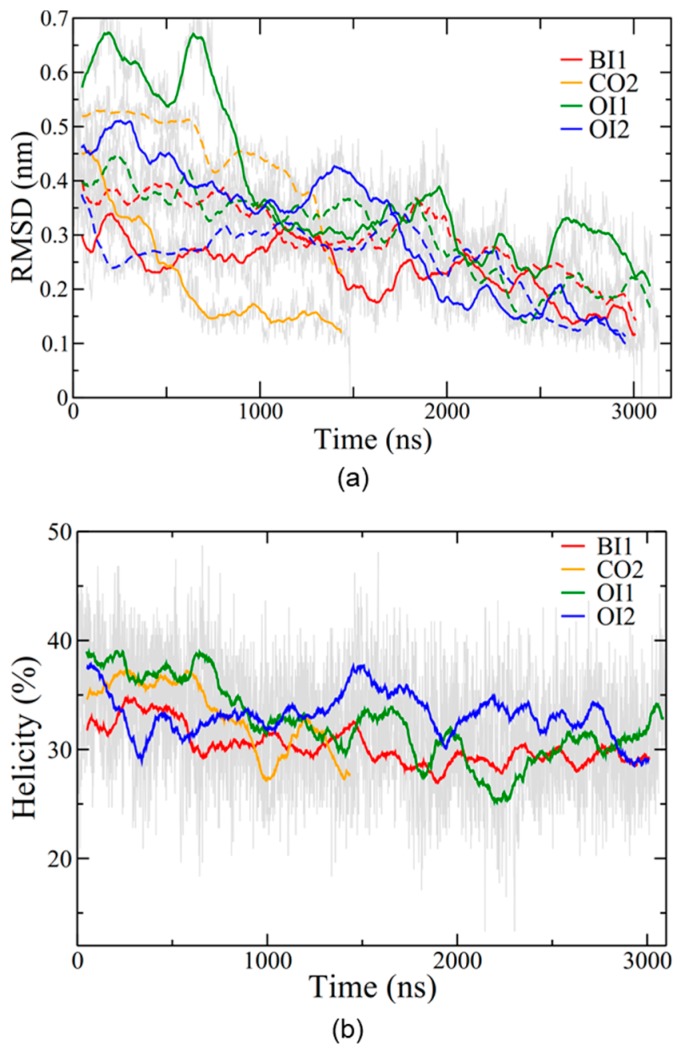
SP-B structure evolution during simulations. (**a**) RMSD (root mean square deviation) for the backbone of the protein as compared to the final structure. RMSD was calculated for each subunit separately, with solid lines for the red subunit (depicted in Figure 2) and dashed lines for the blue subunit. (**b**) Percent helicity of the protein calculated using the program STRIDE [47]. A residue was considered helical if STRIDE labelled it as in an α-helix, 3_10_ helix, or π-helix. The percentage of helical residues at each time step is plotted. Coloured lines represent running averages over 50 points (thus RMSDs do not go to exactly zero at the final time point). Grey lines represent data from individual time steps.

**Figure 5 ijms-20-03863-f005:**
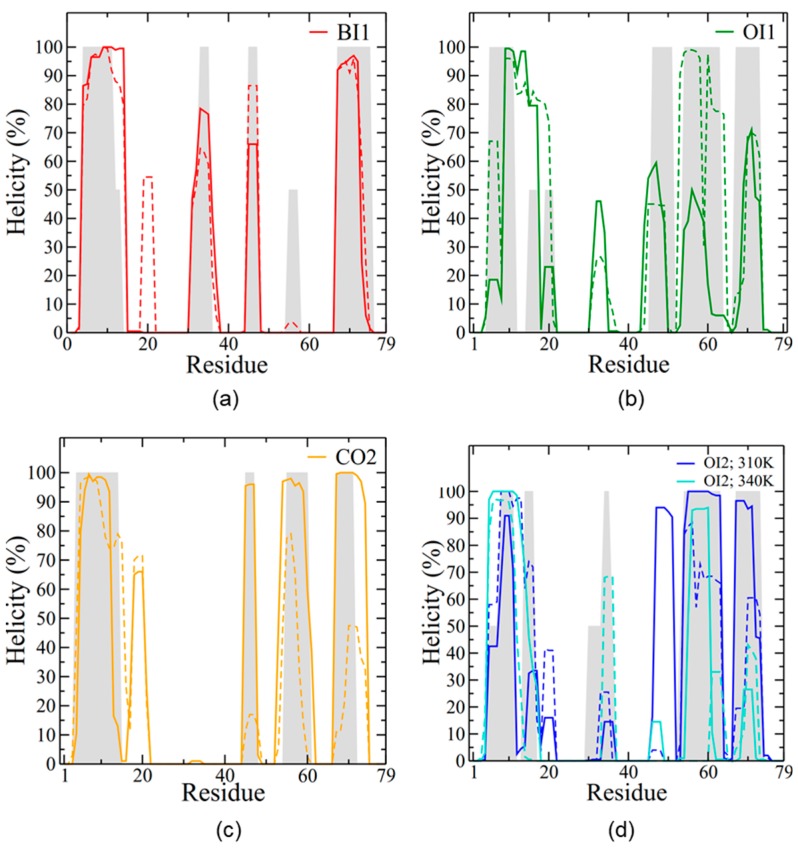
SP-B secondary structure during the last 0.5 μs of simulation. The *y*-axis plots the percentage of time where a residue was in a helical conformation, with frames spaced 2 ns apart. The helicity per residue is reported for (**a**) BI1, (**b**) OI1, (**c**) CO2, and (**d**) OI2. Helicity is calculated for each subunit separately, with solid lines for the red subunit (depicted in Figure 2) and dashed lines for the blue subunit. In (**d**) dark blue represents the OI2 simulation run at a temperature of 310 K and light blue represents the system run at 340 K. The initial helicity is shown with grey shading, averaged for the two subunits.

**Figure 6 ijms-20-03863-f006:**
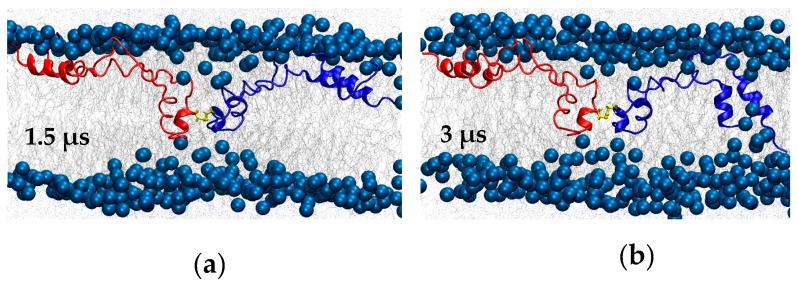
BI1 system snapshots at (**a**) 1.5 μs and (**b**) 3 μs. C48, which forms the inter-subunit disulfide bond, is shown in yellow.

**Figure 7 ijms-20-03863-f007:**
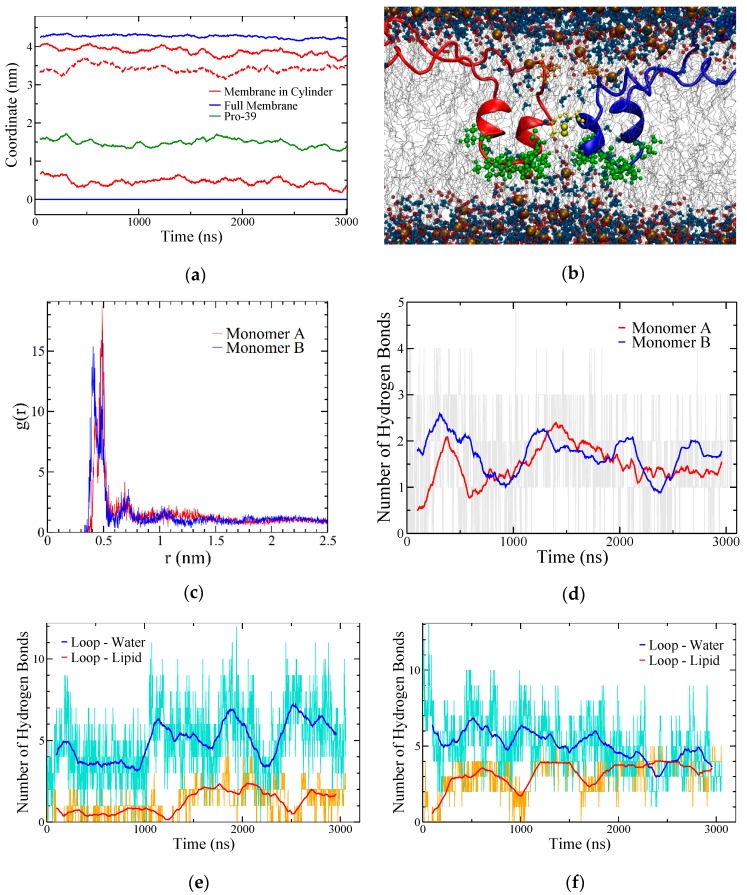
SP-B–lipid interactions in the BI1 system. (**a**) Coordinate plot illustrating bilayer thickness overall (blue) and within a cylinder of 2 nm radius around C48 (red). The maximum density positions of the phosphorous atoms in POPC are shown, with all coordinates referenced to the overall bottom leaflet position. The dotted red line corresponds to the difference between the red lines (i.e., bilayer thickness near the protein). The average position of the two P39 residues is also shown (green) as a measure of the position of the central loops. The data are running averages over 50 points. (**b**) Snapshot at 2.0 μs with R52 sidechain heavy atoms shown as orange balls and sticks, the central loop region from residues 37 to 45 shown in green, and C48, which links the subunits, in yellow. The phosphorous and oxygen atoms in the lipids are shown as brown and red spheres, respectively, and water molecules are shown as blue balls and sticks. (**c**) Radial distribution function of phosphorus atoms in lipids to the terminal side-chain carbon in R52 averaged over the last half of the simulation. (**d**) Time evolution of hydrogen bonding from both R52 residues to POPC headgroups. Grey lines show unaveraged data. (**e**,**f**) Hydrogen bonding between the central loop region of SP-B (residues 37–45) and lipid (orange) and water (cyan), plotted separately for each SP-B subunit. In (**d**–**f**) the running averages are over 100 points.

**Figure 8 ijms-20-03863-f008:**
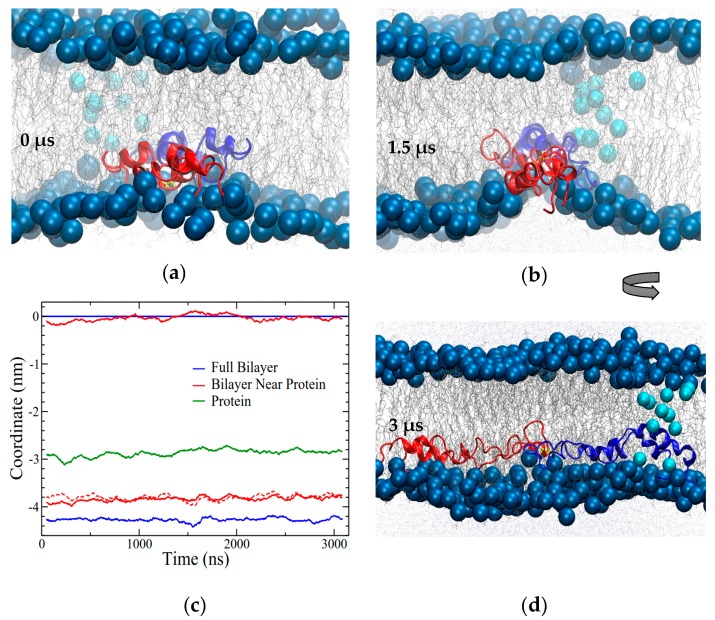
In the OI1 simulation SP-B promoted a crease in the bilayer and also interacted with the preformed pore. The (**a**) initial system configuration and (**b**) configuration at 1.5 μs, looking along the long axis of the protein. The lipid headgroups in the preformed pore are shown with cyan (lighter) blue to help differentiate them from the lipids in the SP-B-induced crease. (**c**) Maximum density positions of phosphorous atoms from lipid headgroups in the bilayer overall (blue) and within 0.5 nm lateral distance from the protein (red), illustrating bilayer thickness. All coordinates are in reference to the overall top leaflet position. The dashed line corresponds to the difference between the top and bottom solid red lines. Green shows the position of maximum density of the protein. The data are running averages over 50 points. (**d**) Final system configuration at 3.135 μs, with the view rotated 90° from (b). The blue subunit contacted the preformed pore at approximately 2 μs.

**Figure 9 ijms-20-03863-f009:**
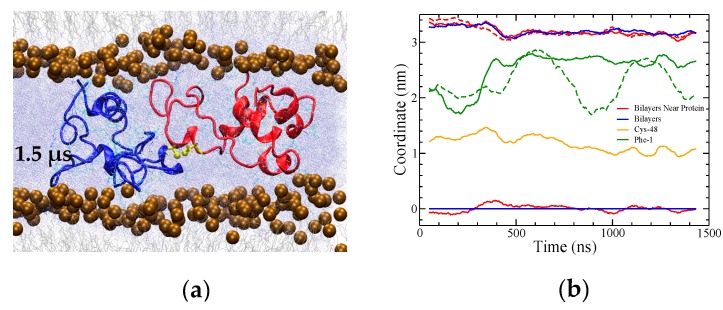
(**a**) Snapshot of the final CO2 system after 1.5 μs of molecular dynamics. The snapshot shows a 7 nm portion of the 11 nm width of the simulation box. (**b**) The distance (difference in *z* coordinates) between the bilayers reduced only very slightly over the course of the simulation. Maximum density positions of phosphorous atoms from lipid headgroups in the bilayers overall (blue) and within 0.5 nm of lateral distance from the protein (red) are plotted, illustrating bilayer separation. All coordinates are in reference to the overall lower leaflet position. The dashed red line corresponds to the difference between the red lines. Green lines show the position of F1, solid for the red subunit and dashed for the blue, as an indicator of the position of the residue 1–7 insertion sequence. The data are running averages over 50 points.

**Figure 10 ijms-20-03863-f010:**
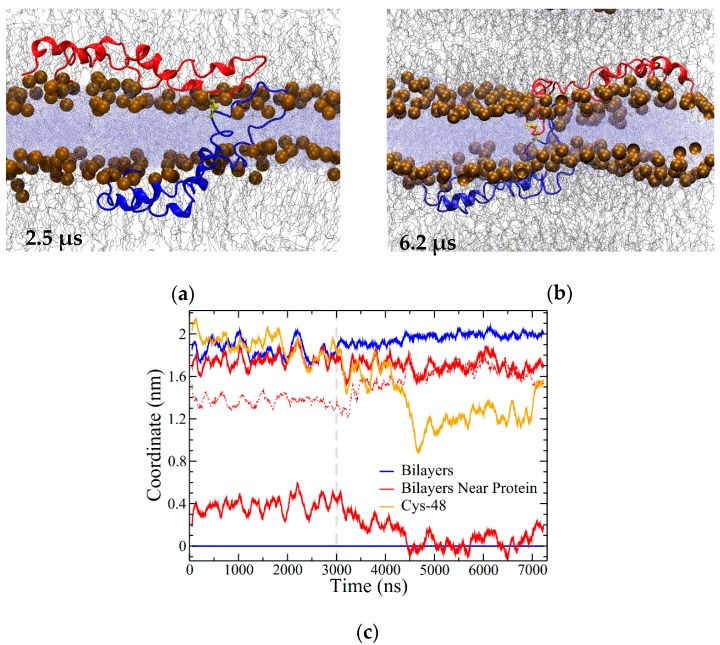
Open In Two Bilayers (OI2) system (**a**) snapshot at 2.5 μs; (**b**) snapshot at 6.2 μs (3.2 μs into the high-temperature portion of the simulation). (**c**) Coordinate plot illustrating the distance between the bilayers around the protein and overall. Coordinates were measured using the highest-density positions of phosphorus atoms in the lipid headgroups, as described in the Methods section. The position of the lower leaflet was set as zero, with all other positions in reference to it. Blue lines illustrate the bilayer separation overall. Red lines show the bilayer positions within 1 nm of the portion of SP-B in the water layer. The dashed red line corresponds to the difference between the red lines (i.e., the bilayer separation). The location of C48 is shown in orange. The dashed grey line corresponds to where temperature was increased from 310 to 340 K. The positions have a resolution of approximately 0.07 nm. Running averages are over 100 points, with 2 ns between points.

**Table 1 ijms-20-03863-t001:** Brief summary of simulations. POPC: 1-palmitoyl-2-oleoyl-sn-glycero-3-phosphocholine. TIP4P: transferable intermolecular potential with 4 points.

System	System Content			Time (μs)	Behavior/Notes
	Lipid (POPC)	Solvent (TIP4P)	Ions (Cl^−^)		
Bent In One Bilayer (BI1)	450	21,000	14	3.059	Central loop region induces a defect N/C-terminal helix pair of one subunit encounters preformed pore and transitions to a transmembrane orientation
Open In One Bilayer (OI1)	450	21,000	14	3.135	SP-B induces a bilayer crease N/C-terminal helix pair of one subunit encounters preformed pore and tilts to ~45°
Closed Out Two Bilayers (CO2)	844	24,917	14	1.482	SP-B does not affect structure of bilayers Very slight reduction in the distance between bilayers
Open In Two Bilayers (OI2)	798	17,633	14	3.002 + 4.274	Temperature increased after 3 μs SP-B promotes close contacts between bilayers Central loop explores headgroup region of opposing bilayer

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
