# Peer review of "All-Atom Molecular Dynamics Simulations of Dimeric Lung Surfactant Protein B in Lipid Multilayers"

_ijms, 2019, doi:10.3390/ijms20163863_

Round 1

Reviewer 1 Report

Robichaud et al. present all-atom MD simulations of the SP-B dimer in the presence of different lipid bilayer configurations. This study is an extension of their previous work on the SP-B monomer (ref 36). Overall, the MS is well presented and interesting behaviours of the protein dimer were observed.

Main comments:

1)    The study uses four starting orientations that were based on the earlier monomer work. It is not clear to me that that end state of the monomer runs are relevant starting points for constructions the dimers. Please include a justification in the MS.

2)    The starting points for the four simulations have a big effect on each run. Notably, none of them converge to similar configurations indicating that the results are highly influenced by the choices of the initial states.  As a result, the four starting points produce four different “stories”.   If additional starting configurations would have been included, presumably more behaviours would have been observed. Please include a discussion of these points in the MS. Can some aspects of the four runs be merged into a more cohesive story?

3)    Please include a section that compares the new results with previously published CG studies (refs 43-45).  The time scales are different, but are some of the observed states (qualitatively) related? Even if there is no apparent correspondence, it would be helpful to point this out.

Minor comments/corrections:

3) Ref 43 is incomplete. 

4)    Fig 3: The range on the Y-axes for panels a, b and c are 0.008, 0.015 and 0.0125 x10^6 kJ/mol, respectively. Also, simulations BI1 and OI1 are combined in one panel, presumably because they start at the same absolute potential value. This makes it difficult to compare the four simulations.  I suggest presenting four panels (one for each run), all with the same Y range.

5)    Fig 5: It would be better to show a single residue range on the X-axis of panels a-d (1-79; and not 1-17,1-79 side by side). Maybe with solid/dashed lines for the two subunits? (unaveraged in all cases). 

Reviewer 2 Report

In this work, the authors describe a series of 4 MD simulations comprising SP-B dimers in interaction with one or two POPC bilayers. The paper is generally well written with just a few misspells or very minor typographical errors.

The science described is also well-grounded and without any serious flaws. That said, I think the draft could be improved and I have a few remarks/questions that could help the authors.

Questions / remarks:

    1. It could be useful to add the amino acid sequence of SP-B rather than having to open another ref to get it. Maybe the authors could mix figure 1 from this work and figure 1 from their previous work (ref #36) to give the reader insights on both the primary and the secondary structure of SP-B.

    2. Given that the authors emphasize the hydrophobicity of SP-B in their introduction. The rationale behind the "CO2" and the "OI2" starting configurations is hard to follow and should be explicated.

    3.1 Is using a preformed pore really vital for the lipid relaxation? Would not be sufficient to let the lipids relax while the protein being restrained for a couple of hundreds of ns and check if eg order parameters, area per lipid or thickness (away from the protein) have reasonable values? The worst-case scenario being that it might be necessary to add or remove a few lipids to lead to correct values but once equilibrated, one would not have to worry about the potentially artifactual effect of a preformed pore (as in the case of the OI1 system).

    3.2 Although using a pore was described in their previous work (ref #36), a snapshot (in supp. data) of it could be beneficial for readers who would like to use this "trick" to their systems.

    4. How is the helicity measured (eg figure 5)? Authors only specify using gromacs tools but no tool provided by gromacs can plot time-averaged helicity against residue id. I can imagine that "gmx do_dssp" (which is more or less a proxy for DSSP) could be part of the toolchain but could not but the sole element. The authors should explicit in the "methods" section how they assign the "helicity" ("H" assignment from DSSP?).

    5. Captions from all "helicity" figures should read what "100 % helicity" means as it may be different from one figure to another (eg percentages in fig 4 have obviously different meanings than in fig 5)

    6. Figure 7.a: I simply do not understand how to properly read this figure and thus understand the interpretation made by the authors. Caption reads that it represents bilayer (overall and around C48) but the actual figure seems to represent Z coordinates (solid lines - also stated in the caption) mixed by thickness values (red dotted line). This is confusing and the figure should be corrected to be clearer. For instance, an alternative could be to draw a map. This can be done using specific tools like apl@voro or fatslim (see https://pythonhosted.org/fatslim/documentation/tutorials.html#tutorial-4-extracting-raw-apl-data-for-further-processing for an APL-based map). The same applies to figure 8.c.

    7. Figures 9.b and 10.c: similarly to fig 7.a, if this figure plots Z coordinates then the caption should not read distance and if the figure represents distances the Y-axis label should not read "coordinates". It is confusing and should be clarified.

    8. As seen in figure 3 and stated by the authors, by the end of the simulations, potential energy for the four systems barely reaches equilibrium and/or has not converged yet (system OI2) which is the rationale behind the heating of system OI2. Yet, it would have been more coherent to do the same to other systems, at least to confirm that equilibrium was reached. Even though 310K is far from the gel-fluid transition temperature of POPC, lipid dynamics at 340K is higher so how sure are the authors that nothing would happen if eg OI1 was heated up to 340K?

    9. The role of the charged residues is clearly an essential part of the discussion as having 14 charges inside a bilayer is not something obvious. Unfortunately, the results section evokes these residues only in the case of OI2 and CO2 ie when charged residues can reach water molecules. But their behavior when charges are "forced" to be a hydrophobic medium (OI1/BI1) and is not really described and hardly discussed.

    10. In their conclusion, the authors claim that SP-B dimer promotes the stabilization or preformed lipid pores but no evidence of such effect is discussed. This assertion should be toned down or compelling evidence should be presented.

Typos:

     - line 123: "course-grained" -> "coarse-grained"

     - line 157-158: font alternates between serif and non serif (copy/paste without removing formatting?)

     - ref 43 lacks issue/volume number and pages. Here is the DOI: 10.1016/j.bbamem.2016.02.030

     - line 190: "every time point" -> "every time step" or "every frame" ?

Round 2

Reviewer 2 Report

The corrections made by the authors improve the quality of their draft without a doubt. In particular, the few things that were not clear in the first version of the draft are now well described.

Very few minor things are still present and should be fixed prior publication:

- a few "time points" are still present in some figure captions

- Figure 7.d: the grey line is not described. I would assume that this line is actually two lines as there are two monomers (and the other lines are running averages). In any case, there is no mention of any grey line in the legend nor in the caption.

Author Response

The corrections made by the authors improve the quality of their draft without a doubt. In particular, the few things that were not clear in the first version of the draft are now well described.

Very few minor things are still present and should be fixed prior publication:

- a few "time points" are still present in some figure captions

Fixed

- Figure 7.d: the grey line is not described. I would assume that this line is actually two lines as there are two monomers (and the other lines are running averages). In any case, there is no mention of any grey line in the legend nor in the caption.

Now mentioned in the legend